# Association of obstructive sleep apnoea with the risk of vascular outcomes and all-cause mortality: a meta-analysis

Chengjuan Xie,[1] Ruolin Zhu,[1] Yanghua Tian,[1,2] Kai Wang[1,2,3]

[1]Department of Neurology, The First Affiliated Hospital of Anhui Medical University, Hefei, China
[2]Collaborative Innovation Centre of Neuropsychiatric Disorders and Mental Health, Anhui Province, Hefei, China
[3]Department of Medical Psychology, Anhui Medical University, Hefei, China

**Correspondence to**
Dr Kai Wang;
wangkai1964@126.com

## ABSTRACT

**Objective** This study aimed to conduct a meta-analysis to explore and summarise the evidence regarding the association between obstructive sleep apnoea (OSA) and the subsequent risk of vascular outcomes and all-cause mortality.

**Methods** Electronic databases PubMed, Embase and the Cochrane Library were searched to identify studies conducted through May 2016. Prospective cohort studies that reported effect estimates with 95% CIs of major adverse cardiac events (MACEs), coronary heart disease (CHD), stroke, cardiac death, all-cause mortality and heart failure for different levels versus the lowest level of OSA were included.

**Results** A total of 16 cohort studies reporting data on 24 308 individuals were included. Of these, 11 studies reported healthy participants, and the remaining five studies reported participants with different diseases. Severe OSA was associated with an increased risk of MACEs (relative risk (RR): 2.04; 95% CI 1.56 to 2.66; P<0.001), CHD (RR: 1.63; 95% CI 1.18 to 2.26; P=0.003), stroke (RR: 2.15; 95% CI 1.42 to 3.24; P<0.001), cardiac death (RR: 2.96; 95% CI 1.45 to 6.01; P=0.003) and all-cause mortality (RR: 1.54; 95% CI 1.21 to 1.97; P<0.001). Moderate OSA was also significantly associated with increased risk of MACEs (RR: 1.16; 95% CI 1.01 to 1.33; P=0.034) and CHD (RR: 1.38; 95% CI 1.04 to 1.83; P=0.026). No significant association was found between mild OSA and the risk of vascular outcomes or all-cause mortality (P>0.05). Finally, no evidence of a factor-specific difference in the risk ratio for MACEs among participants with different levels of OSA compared with those with the lowest level of OSA was found.

**Conclusions** Severe and moderate OSAs were associated with an increased risk of vascular outcomes and all-cause mortality. This relationship might differ between genders. Therefore, further large-scale prospective studies are needed to verify this difference.

## Strengths and limitations of this study:

► This was a meta-analysis to elucidate the association of obstructive sleep apnoea (OSA) with fatal and non-fatal cardiovascular diseases, using a broad search strategy and predefined selection criteria and with no restriction of language or publication status.

► The methodological quality of each study was assessed using the Newcastle-Ottawa Scale for prospective observational studies, and a meta-analysis, sensitivity analysis, subgroup analysis and bias assessment were also conducted.

► Only prospective studies were included, eliminating selection and recall bias that could be of concern in retrospective case–control studies.

► Summary relative risks were calculated to evaluate any potential difference between subsets according to the characteristics of the participants.

► Different cut-off values for the apnoea–hypopnoea index might affect the relationship between OSA and vascular outcomes.

fragmentation.[2] Previous studies suggested that OSA was associated with an increased risk of glaucoma, diabetic kidney disease and metabolic syndrome.[3–5] However, data on the association between OSA and the risk of subsequent vascular outcomes and mortality are both limited and inconclusive. Furthermore, whether these relationships differ according to the characteristics of patients with OSA also needs to be verified.

Several meta-analyses have illustrated that continuous positive airway pressure (CPAP) interventions aimed at OSA may reduce the risk of cardiovascular outcomes. Kim *et al*[6] showed that CPAP treatment for OSA was associated with a lower incidence of stroke and cardiac events. Furthermore, Bratton *et al*[7] indicated that use of both CPAP and mandibular advancement devices was associated with a reduction in the blood pressure among patients with OSA. Nadeem *et al*[8] suggested that CPAP treatment for OSA seemed to improve dyslipidaemia (decrease in total cholesterol and low-density lipoprotein

## INTRODUCTION

Obstructive sleep apnoea (OSA) affects 24% of middle-aged men and 9% of women in the USA, but daytime sleepiness was reported in 17% and 22% of these subjects, respectively.[1] OSA is an increasingly prevalent condition characterised by repetitive obstruction of the upper airway during sleep accompanied by episodic hypoxia, arousal and sleep

and increase in high-density lipoprotein), whereas it did not appear to affect the triglyceride levels. These studies demonstrated that patients with OSA who received interventions had a reduced risk of cardiovascular diseases. Therefore, clarifying the relationship between OSA and vascular outcomes is particularly important as it has not been definitively determined. This study attempted to perform a large-scale examination of the available prospective studies to determine the association of OSA with the potential risk of vascular outcomes and all-cause mortality.

## METHODS

### Data sources, search strategy and selection criteria

This study was conducted and reported according to the Meta-analysis of Observational Studies in Epidemiology protocol (Checklist S1).[9]

Any prospective cohort study that examined the relationship between OSA and vascular outcomes or all-cause mortality was eligible for inclusion into this study, and no restrictions were placed on language or publication status (eg, published, in press or in progress). Electronic databases PubMed, Embase and the Cochrane Library were searched for articles published through May 2016, using the terms 'sleep apnea' OR 'obstructive sleep apneas' AND ('cardiovascular disease' OR 'stroke' OR 'cardiac death' OR 'mortality' OR 'death' OR 'CVD' OR 'myocardial infarction' OR 'coronary events') AND 'clinical trials' AND 'human' as the search terms (online supplementary 1). Manual searches of reference lists were also conducted from all the relevant original and reviewed articles to identify additional eligible studies. The medical subject heading, methods, patient population, design, exposure and outcome variables of these articles were used to identify the relevant studies.

The literature search was independently undertaken by two authors using a standardised approach. Any inconsistencies between these two authors were settled by the primary author until a consensus was reached. The study was eligible for inclusion if the following criteria were met: (1) the study had a prospective cohort design; (2) the study investigated the association between OSA and the risk of major adverse cardiac events (MACEs), coronary heart disease (CHD), stroke, cardiac death, all-cause mortality and heart failure; and (3) the authors reported effect estimates (relative risk (RR), HR or OR) and 95% CIs for comparisons of different levels of OSA versus lowest OSA level. All case–control studies were excluded because various confounding factors could bias the results.

### Data collection and quality assessment

The data collected included the first author's name, publication year, country, sample size, mean age at baseline, percentage of male patients, body mass index (BMI), disease status, assessment of OSA, follow-up duration, effect estimate and its 95% CI, reported endpoints and covariates in the fully adjusted model. For studies that reported several multivariable adjusted RRs, the effect estimate that was maximally adjusted for potential confounders was selected.

The Newcastle-Ottawa Scale (NOS), which is quite comprehensive and has been partially validated for evaluating the quality of observational studies in the meta-analysis, was used to evaluate the methodological quality.[10] The NOS is based on the following three subscales: selection (four items), comparability (one item) and outcome (three items). A 'star system' (range: 0–9) was developed for assessment (table 1). The data extraction and quality assessment were conducted independently by two authors. Information was examined and adjudicated independently by an additional author referring to the original studies.

### Statistical analysis

The relationship between OSA and the risk of vascular outcomes or all-cause mortality based on the effect estimate (OR, RR or HR) and its 95% CI was examined in each study. HR was considered to be equivalent to RR in cohort studies. Given the low incidence of vascular outcomes and all-cause mortality, ORs could be considered as accurate estimates of RRs.[11] A semiparametric method was first used to evaluate the association of mild OSA (apnoea–hypopnea index (AHI): 5–15), moderate OSA (AHI: 15–30) and severe OSA (AHI >30) with the risk of vascular outcomes or all-cause mortality in order to analyse the trend between OSA levels and vascular outcomes or all-cause mortality risk.[12] For each individual study, each category of AHI was reclassified based on its calculated midpoint (for closed categories) or median (for open categories, assuming a normal distribution for AHI). The control category was composed of participants with the lowest AHI or normal participants in that study. Furthermore, when an individual study provided more than one median AHI level for classification among the three categories (ie, mild, moderate or severe OSA), a fixed-effects model was used to calculate their summary RRs and 95% CIs to obtain effect estimates for each category.[13] If the study data were not broken down by AHI but rather by oxygen desaturation index, classification into the OSA categories was carried out based on the judgement of the clinicians. A random-effects model was then used to calculate summary RRs and 95% CIs for mild, moderate and severe OSA versus normal.[14] Finally, the ratio of RRs and the corresponding 95% CIs between subgroups were estimated using specific RRs and 95% CIs in each group based on the country, mean age, gender, BMI, disease status and duration of the follow-up period.[15]

Heterogeneity between studies was investigated using the Q statistic, and P values <0.10 were considered as indicative of significant heterogeneity.[16 17] Subgroup analyses were conducted for mild, moderate and severe OSA and the risk of MACEs based on the country, mean age, gender, BMI, disease status and duration of the follow-up period. A sensitivity analysis was also performed by

**Table 1** Baseline characteristic of studies included in the systematic review and meta-analysis

| Study | Country | Sample size | Mean age | Percentage male (%) | BMI | Disease status | Assessment OSA | AHI or ODI categories | Follow-up duration (year) | Reported outcomes | Adjusted factors | NOS score |
|---|---|---|---|---|---|---|---|---|---|---|---|---|
| Mooe et al 2000[21] | Sweden | 408 | 59.1 | 58.4 | 27.0 | CAD | Limited PSG | <5; 5–10; 10–15; ≥15 | 5.1 | CHD, stroke, all-cause mortality | Age, sex, BMI, hypertension, DM, LVF and coronary intervention | 7 |
| Gottlieb et al 2010[22] | USA | 4422 | 62.4 | 43.5 | 28.2 | Healthy | PSG | <5; 5–15; 15–30; ≥30 | 8.7 | HF | Age, race, BMI, smoking, DM, SBP, DBP, TC, HDL-C, lipid-lowering medications and antihypertensive medications | 8 |
| Campos-Rodriguez et al 2012[23] | Spain | 1116 | 56.1 | 0.0 | 36.6 | Healthy | PSG | <10; 10–29; ≥30 | 6.0 | Cardiac death | Age, BMI, DM, hypertension and previous CVD | 8 |
| Marin et al 2005[24] | Spain | 1729 | 49.9 | 100 | 28.7 | Healthy | PSG | 5–30; ≥30 | 10.1 | Cardiac death and CHD | Age, diagnostic group, presence of CVD, DM, hypertension, lipid disorders, smoking, alcohol, SBP DBP, blood glucose, TC, TG and use of antihypertensive, lipid-lowering and antidiabetic drugs | 9 |
| Young et al 2008[25] | USA | 1522 | 48.0 | 55.0 | 28.6 | Healthy | PSG | 5–15; 15–30; ≥30 | 18.0 | Cardiac death, all-cause mortality and CHD | Age, age-squared, sex, BMI and BMI squared | 8 |
| Redline et al 2010[26] | USA | 5422 | 62.9 | 45.4 | 27.8 | Healthy | PSG | Quartile I (0–4.05); quartile II (4.05–9.50); quartile III (9.50–19.13); quartile IV (>19.13) | 8.7 | Stroke | Age, BMI, race, smoking, SBP, DM and antihypertensive medications | 8 |
| Arzt et al 2005[27] | USA | 1189 | 47.0 | 55.0 | 30.0 | Healthy | PSG | <5; 5–20; ≥20 | 4.0 | Stroke | Age, sex, and BMI | 7 |
| Punjabi et al 2008[28] | USA | 6294 | 62.5 | 47.0 | 27.8 | Healthy | PSG | Quartile I (0–8.50); quartile II (8.51–15.09); quartile III (15.10–24.28); quartile IV (>24.28) | 8.2 | CHD, all-cause mortality | Age, sex, race, BMI, SBP, DBP, smoking, prevalent hypertension, DM and CVD | 8 |
| Shah et al 2010[29] | USA | 1436 | 59.7 | 69.4 | 32.9 | Healthy | PSG | <5; 5–14; 15–29; ≥30 | 2.9 | CHD, cardiac death | Age, race, sex, smoking, alcohol, BMI, AF, DM, hypertension and hyperlipidaemia | 7 |
| Yaggi et al 2005[30] | USA | 1022 | 60.2 | 71.3 | 32.8 | Healthy | PSG | ≤3; 4–12; 13–36; ≥36 | 3.4 | Stroke and all-cause mortality | Age, sex, race, smoking, alcohol, BMI, DM, hyperlipidaemia, AF and hypertension | 8 |

Continued

**Table 1** Continued

| Study | Country | Sample size | Mean age | Percentage male (%) | BMI | Disease status | Assessment OSA | AHI or ODI categories | Follow-up duration (year) | Reported outcomes | Adjusted factors | NOS score |
|---|---|---|---|---|---|---|---|---|---|---|---|---|
| Martinez-Garcia et al 2009[31] | Spain | 166 | 73.3 | 59.0 | 28.1 | Ischaemic stroke | PSG | 0–9; 10–19; ≥20 | 5.0 | All-cause mortality | Age, sex, Barthel index, AHI and CPAP treatment groups, previous stroke or TIA, diabetes, hypercholesterolaemia, BMI, smoking, arterial hypertension, atrial fibrillation, significant carotid stenosis and fibrinogen levels | 7 |
| Munoz et al 2006[32] | Spain | 1034 | 79.8 | 57.0 | 26.8 | Healthy | PSG | <30; ≥30 | 6.0 | Stroke | Sex | 7 |
| Leão et al 2016[33] | Portugal | 73 | 62.4 | 75.0 | 27.6 | Acute coronary syndrome | PSG | 5–15; 15–30; ≥30 | 6.3 | CHD | Sex | 7 |
| Fornadi et al 2014[34] | Hungary | 100 | 51.0 | 56.8 | 26.8 | Kidney transplant recipients | PSG | 5–15; 15–30; ≥30 | 6.3 | All-cause mortality | Unadjusted | 6 |
| Kenzerska et al 2014[35] | Canada | 10149 | 49.9 | 62.0 | 30.1 | Healthy | PSG | <5; 5–15; 15–30; ≥30 | 5.7 | All-cause mortality | Traditional CV risk factors | 7 |
| Won et al 2013[36] | USA | 281 | 65.0 | 98.0 | 34.0 | Ischaemic heart disease and myocardial injury | PSG | 5–30; ≥30 | 4.1 | All-cause mortality | NA | 6 |

AF, atrial fibrillation; AHI, apnoea–hypopnea index; BMI, body mass index; CAD, coronary artery disease; CHD, coronary heart disease; CPAP, continuous positive airway pressure; CV, cardiovascular; CVD, cardiovascular disease; DBP, diastolic blood pressure; DM, diabetes mellitus; HDL-C, high-density lipoprotein cholesterol; HF, heart failure; LVF, left ventricular function; NA, not applicable; NOS, Newcastle–Ottawa Scale; ODI, oxygen desaturation index; OSA, obstructive sleep apnoea; PSG, polysomnography; SBP, systolic blood pressure; TC, total cholesterol; TG, triglyceride; TIA, transient ischaemic attack.

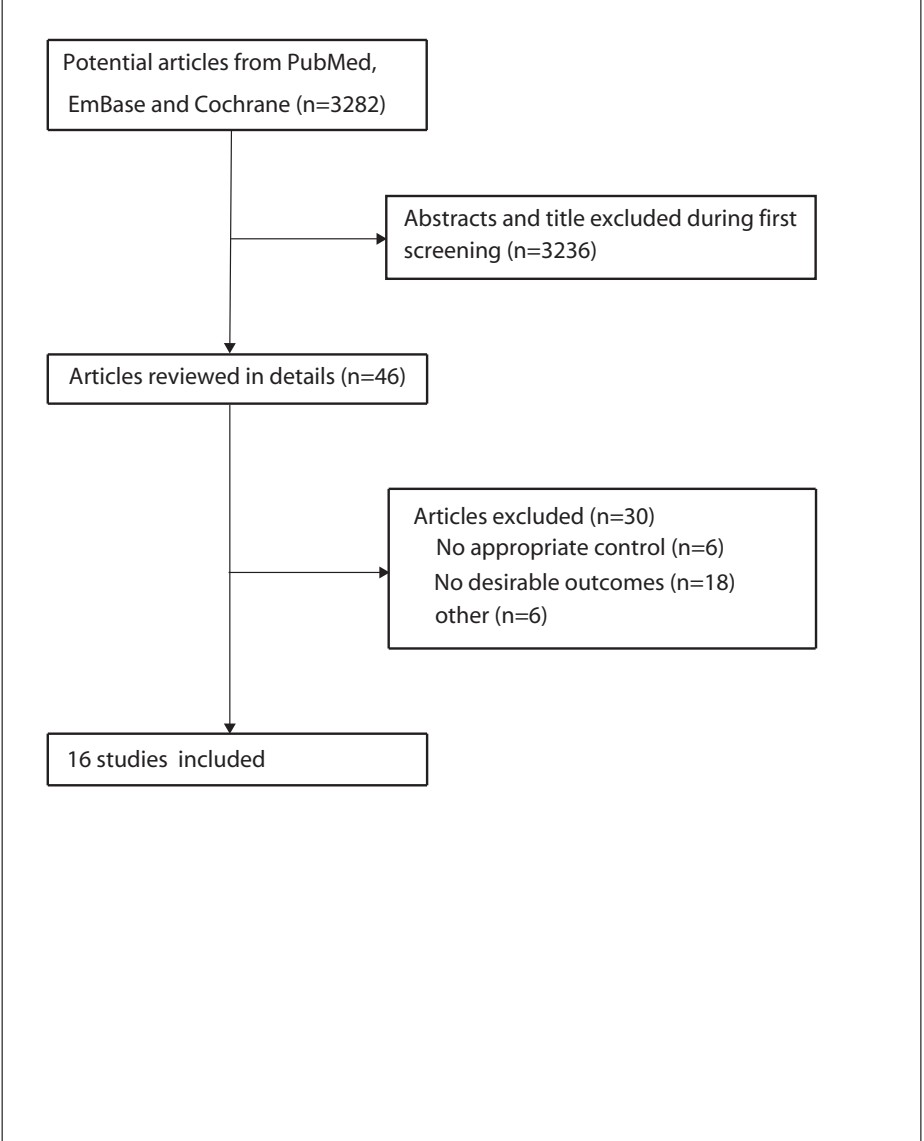

**Figure 1** Study selection process.

removing each individual study from the meta-analysis.[18] Several methods were used to check for potential publication bias. Visual inspections of funnel plots for MACEs were conducted. The Egger[19] and Begg[20] tests were also used to statistically assess publication bias for MACEs. All reported P values were two sided, and P values <0.05 were regarded as statistically significant for all included studies. Statistical analyses were performed using the STATA software (V.12.0).

## RESULTS
### Literature search
The results of the study selection process are shown in figure 1. An initial electronic search yielded 3282 articles, of which 3236 duplicates and irrelevant studies were excluded, and 46 potentially eligible studies were selected. After detailed evaluations, 16 prospective studies were selected for the final meta-analysis.[21–36]

No new studies qualified for inclusion after a manual search of the reference lists of these studies. The general characteristics of the included studies are presented in table 1.

### Study characteristics
A total of 16 studies with 24 308 individuals qualified for this study. The follow-up period for participants was 2.9–18.0 years, while 73–10 149 individuals were included in each study. Eight studies were conducted in the USA, four in Spain, one in Sweden, one in Portugal, one in Hungary and one in Canada. Furthermore, 11 studies reported healthy participants, and the remaining five studies reported participants with different diseases. The mean BMI ranged from 26.8 to 34.0 kg/m². Fourteen studies used polysomnography (PSG), and the remaining one study used limited PSG to assess the levels of OSA. The study quality was assessed using the NOS (table 1). Overall, one study had a score of 9, six studies had a score

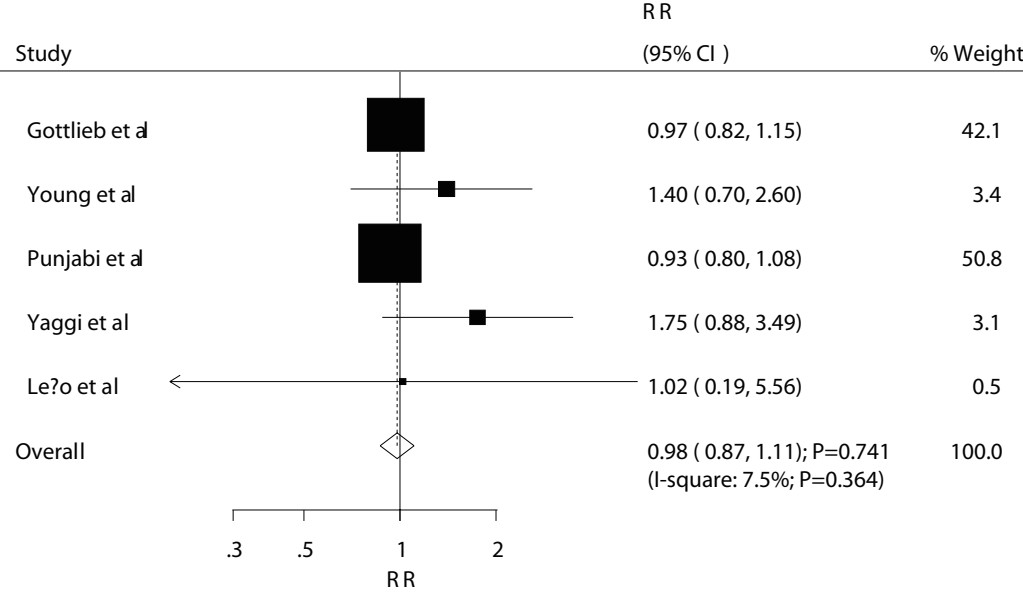

**Figure 2** Association between mild OSA and MACEs. MACES, major adverse cardiac events; OSA, obstructive sleep apnoea; RR, relative risk.

of 8, seven studies had a score of 7 and the remaining two studies had a score of 6.

### OSA and MACE risk

The summary RRs showed that mild OSA was not associated with MACEs (RR: 0.98; 95% CI 0.87 to 1.11; P=0.741; figure 2 and table 2). Furthermore, the pooled analysis results for moderate and severe OSA indicated that they had a harmful effect on the risk of MACEs (moderate: RR: 1.16; 95% CI 1.01 to 1.33; P=0.034; figure 3 and table 2; severe: RR: 2.04; 95% CI 1.56 to 2.66; P<0.001; figure 4 and table 2). A subgroup analysis for MACEs was conducted to minimise heterogeneity among the included studies and evaluate the relationship between OSA and MACEs in specific subpopulations (table 3). Overall, participants with moderate OSA were associated with an increased risk of MACEs if individuals did not have other diseases (RR: 1.16; 95% CI 1.01 to 1.33; P=0.034). Furthermore, no significant association was found between severe OSA and MACEs if the study included only women (RR: 1.98; 95% CI 0.64 to 6.06; P=0.234); in other subsets, severe OSA was associated with an increased risk of MACEs (table 3). Finally, no evidence of a factor-specific difference was

found in the RR for MACEs among participants with OSA compared with controls (table 3).

### OSA and CHD risk

The pooled data of meta-analysis showed that mild OSA was not associated with the risk of CHD (RR: 1.25; 95% CI 0.95 to 1.66; P=0.117; table 2 and online supplementary 2), whereas moderate OSA (RR: 1.38; 95% CI 1.04 to 1.83; P=0.026; table 2 and online supplementary 2) and severe OSA (RR: 1.63; 95% CI 1.18 to 2.26; P=0.003; table 2 and online supplementary 2) were associated with a significantly increased risk of CHD. Stratified analyses according to gender were conducted for different levels of OSA versus normal group, and it was found that patients with severe OSA had significantly increased the risk of CHD in men (RR: 1.65; 95% CI 1.06 to 2.57; P=0.027). No other significant differences were detected (table 4).

### OSA and stroke risk

Pooled analysis results indicated no association between mild OSA (RR: 1.29; 95% CI 0.69 to 2.41; P=0.424; table 2 and online supplementary 2) and moderate OSA (RR: 1.35; 95% CI 0.82 to 2.23; P=0.245; table 2 andonline

**Table 2** Summary of the relative risks of all outcomes evaluated

| Outcomes | Mild OSA (RR with 95% CI) | P value for mild OSA | Moderate OSA (RR with 95% CI) | P value for moderate OSA | Severe OSA (RR with 95% CI) | P value for severe OSA |
|---|---|---|---|---|---|---|
| MACEs | 0.98 (0.87 to 1.11) | 0.741 | 1.16 (1.01 to 1.33) | 0.034 | 2.04 (1.56 to 2.66) | <0.001 |
| CHD | 1.25 (0.95 to 1.66) | 0.117 | 1.38 (1.04 to 1.83) | 0.026 | 1.63 (1.18 to 2.26) | 0.003 |
| Stroke | 1.29 (0.69 to 2.41) | 0.424 | 1.35 (0.82 to 2.23) | 0.245 | 2.15 (1.42 to 3.24) | <0.001 |
| Cardiac death | 1.80 (0.68 to 4.76) | 0.236 | 1.11 (0.53 to 2.35) | 0.781 | 2.96 (1.45 to 6.01) | 0.003 |
| All-cause mortality | 1.26 (0.77 to 2.07) | 0.354 | 1.04 (0.60 to 1.79) | 0.895 | 1.54 (1.21 to 1.97) | <0.001 |
| Heart failure | 1.02 (0.78 to 1.34) | 0.868 | 1.07 (0.74 to 1.54) | 0.719 | 1.44 (0.94 to 2.21) | 0.097 |

CHD, coronary heart disease; MACE, major cardiovascular event; OSA, obstructive sleep apnoea; RR, relative risk.

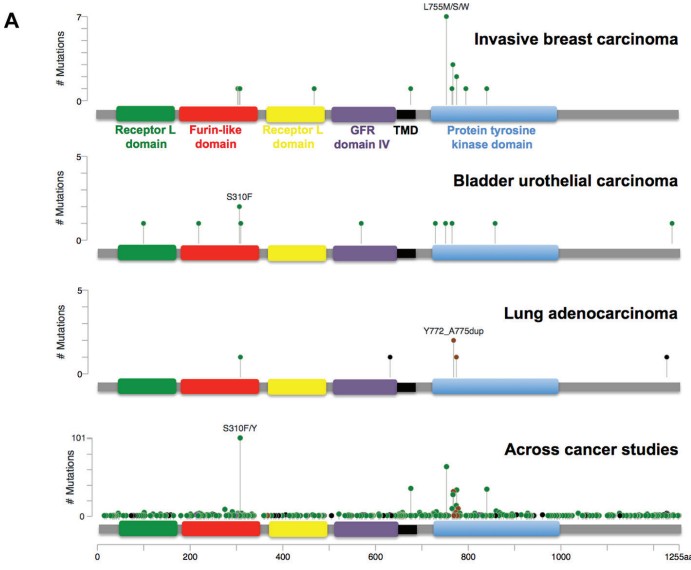

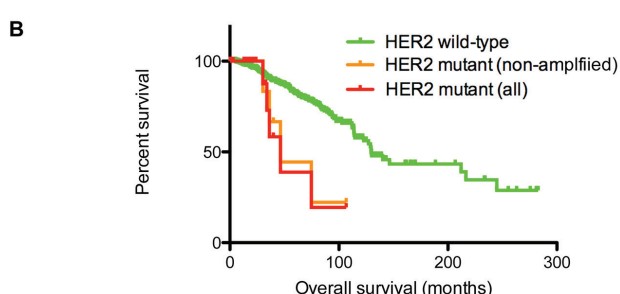

**Figure 3** Association between moderate OSA and MACEs. MACES, major adverse cardiac events; OSA, obstructive sleep apnoea.

supplementary 2) and stroke, whereas severe OSA was associated with an increased risk of stroke (RR: 2.15; 95% CI 1.42 to 3.24; P<0.001; table 2 and online supplementary 2). Subgroup analysis on the basis of gender indicated that severe OSA had a harmful effect on the risk of stroke in men (RR: 2.86; 95% CI 1.10 to 7.41; P=0.031; table 4).

### OSA and cardiac death risk

The summary RRs showed that mild OSA (RR: 1.80; 95% CI 0.68 to 4.76; P=0.236; table 2 and online supplementary 2) and moderate OSA (RR: 1.11; 95% CI 0.53 to 2.35; P=0.781; table 2 and online supplementary 2) were not associated with cardiac death risk, whereas severe OSA significantly increased the risk of cardiac death (RR: 2.96; 95% CI 1.45 to 6.01; P=0.003; table 2 and online supplementary 2). Subgroup analysis showed that severe OSA was associated with an increased risk of cardiac death in men (RR: 2.87; 95% CI 1.13 to 7.27; P=0.026; table 4).

### OSA and all-cause mortality risk

No significant association was found between mild OSA (RR: 1.26; 95% CI 0.77 to 2.07; P=0.354; table 2 and online

supplementary 2), moderate OSA (RR: 1.04; 95% CI 0.60 to 1.79; P=0.895; table 2 and online supplementary 2) and all-cause mortality risk. However, severe OSA had a harmful impact on the all-cause mortality (RR: 1.54; 95% CI 1.21 to 1.97; P<0.001; table 2 and online supplementary 2). Stratified analysis suggested that severe OSA increased the risk of all-cause mortality in men (RR: 1.72; 95% CI 1.22 to 2.43; P=0.002) and women (RR: 3.50; 95% CI 1.23 to 9.97; P=0.019; table 4).

### OSA and heart failure risk

The summary results indicated no significant differences between mild OSA (RR: 1.02; 95% CI 0.78 to 1.34; P=0.868), moderate OSA (RR: 1.07; 95% CI 0.74 to 1.54; P=0.719) and severe OSA (RR: 1.44; 95% CI 0.94 to 2.21; P=0.097) and the risk of heart failure (table 2 and online supplementary 2). Subgroup analysis reported similar results compared with the overall analysis (table 4).

### Publication bias

Review of the funnel plots could not rule out the potential publication bias for MACEs (figure 5). The Egger and Begg test results showed no evidence of publication bias

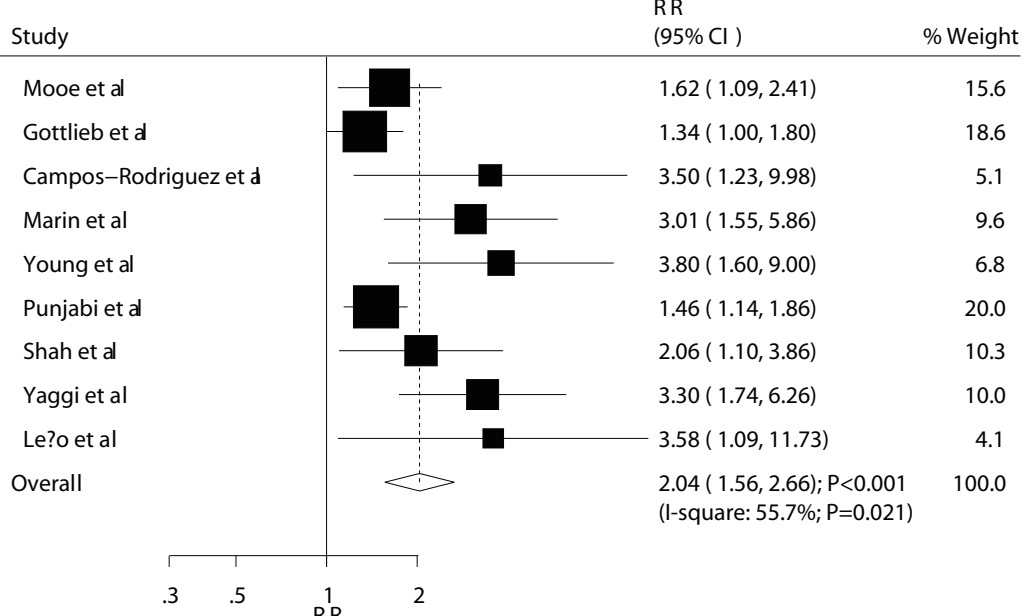

**Figure 4** Association between severe OSA and MACEs. MACES, major adverse cardiac events; OSA, obstructive sleep apnoea; RR, relative risk.

for MACEs of mild OSA (P value for Egger: 0.132; P value for Begg: 0.221) and moderate OSA (P value for Egger: 0.052; P value for Begg: 0.452). Although the Begg test showed no evidence of publication bias for MACEs of severe OSA (P=0.118), the Egger test showed potential evidence of publication bias for MACEs of severe OSA (P<0.001). The conclusion did not change after adjustment for publication bias using the trim-and-fill method.[37]

## DISCUSSION

The present study was based on prospective cohort studies and explored all possible correlations between OSA and the outcomes of MACEs, CHD, stroke, cardiac death, all-cause mortality and heart failure. This large quantitative study included 24 308 individuals from 16 prospective cohort studies with a broad range of populations. The findings from the present meta-analysis suggested that mild OSA had no significant impact on the risk of vascular outcomes and all-cause mortality, moderate OSA was associated with an increased risk of MACEs and CHD and severe OSA had a harmful effect on the risk of MACEs, CHD, stroke, cardiac death and all-cause mortality.

A previous meta-analysis suggested that OSA was associated with stroke, but its relationship with ischaemic heart disease and cardiovascular mortality needs further research.[38] However, this study could not illustrate the impact of different levels of OSA on the risk of serious cardiovascular outcomes. Furthermore, Dong et al[39] suggested that moderate-to-severe OSA significantly increased the risk of cardiovascular diseases, in particular, the risk of stroke. Similarly, Ge et al[40] indicated that severe OSA is a strong independent predictor of cardiovascular and all-cause mortality. CPAP treatment was associated with decreased cardiovascular mortality. However, these

two studies could not evaluate the association of OSA with the risk of vascular outcomes and all-cause mortality in specific subpopulations. In addition, Wang et al[41] suggested that severe OSA significantly increased the risk of CHD and stroke, and all-cause mortality. A positive association with CHD was observed for moderate OSA but not for mild OSA. However, whether this relationship differs according to the characteristics of participants remains unclear. Finally, Xie et al[42] conducted a meta-analysis to evaluate the relationship between OSA and recurrent vascular events and all-cause mortality. However, they just compared the highest AHI versus lowest AHI, whereas the degree of OSA and subsequent adverse outcomes were not available. Therefore, a comprehensive meta-analysis of these prospective cohort studies was performed to evaluate any possible correlates between OSA and vascular outcomes.

No significant difference was observed between mild OSA and the risk of vascular outcomes. However, several studies included in this study reported inconsistent results. Young et al[25] suggested that mild OSA significantly increased the risk of CHD by 92%, whereas Punjabi et al[28] indicated that mild OSA might have a harmful effect on the risk of CHD. This might be because these two studies used healthy individuals as controls, which may make them more susceptible to acquired significant conclusion. Furthermore, most of these studies did not take into account potential confounders for the risk of cardiovascular disease. Moderate-to-severe OSA might play an important role in the risk of vascular outcomes. Shah et al[29] concluded that OSA increased the risk of coronary events or death from cardiovascular causes. Nearly all included studies reported adverse outcomes for severe OSA. Finally, Previous studies indicated that OSA was a

**Table 3** Subgroup analyses for MACEs

| Variable | Subgroup | Mild OSA (RR with 95% CI) | P value for mild OSA | Moderate OSA (RR with 95% CI) | P value for moderate OSA | Severe OSA (RR with 95% CI) | P value for severe OSA |
|---|---|---|---|---|---|---|---|
| Country | USA | 1.00 (0.85 to 1.17) | 0.977 | 1.14 (0.99 to 1.32) | 0.064 | 1.90 (1.35 to 2.67) | <0.001 |
| | Other | 1.02 (0.19 to 5.52) | 0.982 | 1.44 (0.83 to 2.50) | 0.198 | 2.35 (1.52 to 3.65) | <0.001 |
| | USA versus other | 0.98 (0.18 to 5.32)* | 0.982 | 0.79 (0.45 to 1.40)* | 0.422 | 0.81 (0.46 to 1.41)* | 0.453 |
| Mean age | ≥60 | 0.96 (0.86 to 1.08) | 0.540 | 1.13 (0.97 to 1.33) | 0.117 | 1.78 (1.23 to 2.57) | 0.002 |
| | <60 | 1.40 (0.73 to 2.70) | 0.315 | 1.51 (0.94 to 2.41) | 0.086 | 2.31 (1.64 to 3.24) | <0.001 |
| | ≥60 versus <60 | 0.69 (0.35 to 1.33)* | 0.265 | 0.75 (0.46 to 1.23)* | 0.252 | 0.77 (0.47 to 1.27)* | 0.309 |
| Gender | Male | 0.92 (0.73 to 1.15) | 0.455 | 1.10 (0.85 to 1.42) | 0.449 | 1.81 (1.14 to 2.89) | 0.012 |
| | Female | 1.97 (0.47 to 8.25) | 0.353 | 1.36 (0.67 to 2.76) | 0.399 | 1.98 (0.64 to 6.06) | 0.234 |
| | Male versus female | 0.47 (0.11 to 1.99)* | 0.304 | 0.81 (0.38 to 1.72)* | 0.581 | 0.91 (0.27 to 3.08)* | 0.885 |
| BMI | ≥30 | 1.75 (0.88 to 3.49) | 0.111 | 1.70 (0.94 to 3.07) | 0.079 | 2.72 (1.80 to 4.10) | <0.001 |
| | <30 | 0.96 (0.86 to 1.07) | 0.449 | 1.14 (0.99 to 1.31) | 0.078 | 1.80 (1.36 to 2.38) | <0.001 |
| | ≥30 versus <30 | 1.82 (0.91 to 3.66)* | 0.092 | 1.49 (0.81 to 2.74)* | 0.198 | 1.51 (0.92 to 2.49)* | 0.104 |
| Disease statues | Healthy | 1.00 (0.85 to 1.17) | 0.977 | 1.16 (1.01 to 1.33) | 0.034 | 2.12 (1.53 to 2.94) | <0.001 |
| | Other | 1.02 (0.19 to 5.52) | 0.982 | – | – | 1.96 (1.01 to 3.81) | 0.047 |
| | Healthy versus Other | 0.98 (0.18 to 5.32)* | 0.982 | – | – | 1.08 (0.52 to 2.27)* | 0.835 |
| Follow-up duration | ≥6 | 0.96 (0.86 to 1.07) | 0.449 | 1.14 (0.99 to 1.31) | 0.064 | 2.06 (1.43 to 2.95) | <0.001 |
| | <6 | 1.75 (0.88 to 3.49) | 0.111 | 1.74 (0.87 to 3.49) | 0.120 | 2.10 (1.39 to 3.17) | <0.001 |
| | ≥6 versus <6 | 0.55 (0.27 to 1.10)* | 0.092 | 0.66 (0.32 to 1.33)* | 0.242 | 0.98 (0.57 to 1.70)* | 0.945 |

*Reported as ratio of RR and 95% CI.

BMI, body mass index; MACES, major adverse cardiac events; OSA, obstructive sleep apnoea; RR, relative risk.

**Table 4** Gender difference for other outcomes

| Outcome | Subgroup | Mild OSA (RR with 95% CI) | P value for mild OSA | Moderate OSA (RR with 95% CI) | P value for moderate OSA | Severe OSA (RR with 95% CI) | P value for severe OSA |
|---|---|---|---|---|---|---|---|
| CHD | Men | 0.93 (0.72 to1.21) | 0.596 | 1.09 (0.80 to1.48) | 0.582 | 1.65 (1.06 to2.57) | 0.027 |
| | Women | 1.92 (0.43 to 8.64) | 0.394 | 1.51 (0.38 to 5.97) | 0.559 | 1.10 (0.12 to 9.87) | 0.933 |
| | Men versus women | 0.48 (0.11 to 2.22)* | 0.351 | 0.72 (0.18 to 2.96)* | 0.651 | 1.50 (0.16 to 14.22)* | 0.724 |
| Stroke | Men | 1.86 (0.67 to 5.14) | 0.232 | 1.86 (0.70 to 4.95) | 0.214 | 2.86 (1.10 to 7.41) | 0.031 |
| | Women | 1.34 (0.76 to 2.36) | 0.311 | 1.20 (0.67 to 2.15) | 0.542 | 1.21 (0.65 to 2.25) | 0.546 |
| | Men versus women | 1.39 (0.43 to 4.45)* | 0.581 | 1.55 (0.50 to 4.84)* | 0.451 | 2.36 (0.76 to 7.38)* | 0.138 |
| Cardiac death | Men | – | – | 1.15 (0.41 to 3.23) | 0.791 | 2.87 (1.13 to 7.27) | 0.026 |
| | Women | – | – | 0.94 (0.19 to 4.61) | 0.939 | 3.71 (0.41 to 33.87) | 0.245 |
| | Men versus women | – | – | 1.22 (0.18 to 8.17)* | 0.935 | 0.77 (0.07 to 8.49)* | 0.834 |
| All-cause mortality | Men | – | – | – | – | 1.72 (1.22 to 2.43) | 0.002 |
| | Women | – | – | – | – | 3.50 (1.23 to 9.97) | 0.019 |
| | Men versus women | – | – | – | – | 0.49 (0.16 to 1.48)* | 0.206 |
| Heart failure | Men | 0.88 (0.57 to 1.35) | 0.561 | 1.13 (0.68 to 1.88) | 0.639 | 1.58 (0.93 to 2.67) | 0.088 |
| | Women | 1.13 (0.80 to 1.60) | 0.493 | 1.01 (0.60 to 1.70) | 0.970 | 1.19 (0.56 to 2.52) | 0.650 |
| | Men versus women | 0.78 (0.45 to 1.35)* | 0.376 | 1.12 (0.54 to 2.32)* | 0.762 | 1.33 (0.53 to 3.33)* | 0.545 |

*Reported as ratio of RR and 95% CI.
CHD, coronary heart disease; OSA, obstructive sleep apnoea; RR, relative risk.

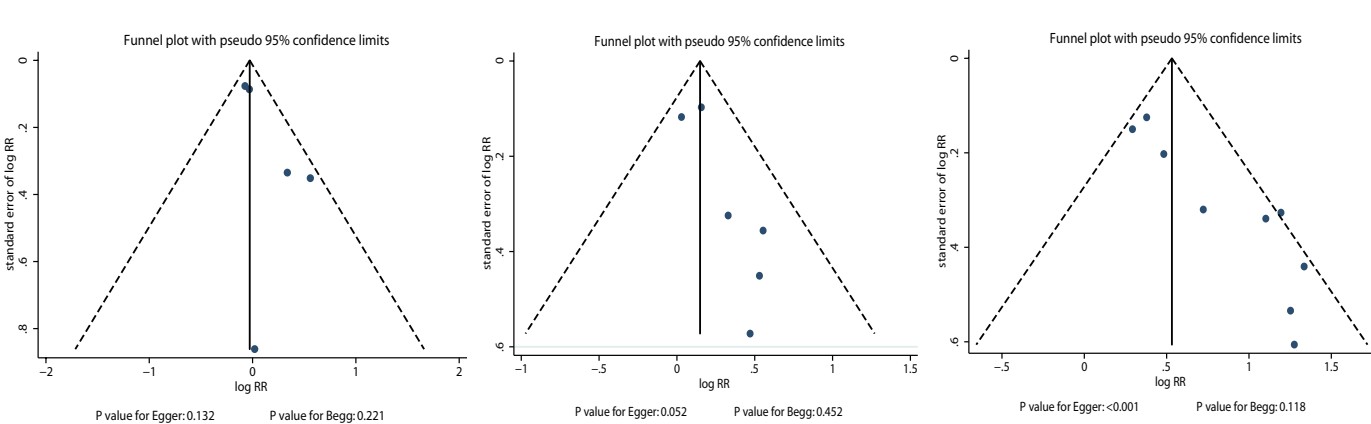

A. mild OSA and MACEs                    B. moderate OSA and MACEs                    C. severe OSA and MACEs

P value for Egger: 0.132    P value for Begg: 0.221

P value for Egger: 0.052    P value for Begg: 0.452

P value for Egger: <0.001    P value for Begg: 0.118

**Figure 5**    Funnel plots. MACES, major adverse cardiac events; OSA, obstructive sleep apnoea; RR, relative risk.

cause of diabetes, which was an independent risk factor for MACEs.[43]

Subgroup analyses reported similar conclusions. Gender might have an impact on the relationship between OSA and CHD, stroke or cardiac death, although the sex difference was not statistically significant. The possible reasons could be the lower prevalence of severe OSA in women and the later age of onset of OSA in women than in men. Furthermore, OSA in women always occurred after menopause. Physiological response to OSA is another reason for this non-significant difference. Finally, these conclusions might be unreliable because smaller cohorts were included in each subset. Therefore, further large-scale studies were needed to verify this difference. Therefore, a relative result was given, and a synthetic and comprehensive review was provided.

No significant difference was found between mild or moderate OSA and all-cause mortality, while severe OSA was associated with an increased risk of all-cause mortality. Furthermore, these significant associations were also observed in men and women separately. Although the effect estimate in women was larger than that in men, no gender difference was found in the relationship between OSA and all-cause mortality. This might be because the number of studies that reported the relationship between severe OSA and all-cause mortality was smaller than expected, and a broad 95% CI was acquired. Therefore, the association of severe OSA with all-cause mortality in women was variable and should be verified in future large-scale prospective studies.

Three strengths of this study should be highlighted. First, only prospective studies were included, which eliminated selection and recall bias, and could be of concern in retrospective case–control studies. Second, the large sample size allowed us to quantitatively assess the association of OSA with the risk of vascular outcomes and mortality, and thus the findings were potentially more robust than those of any individual study. Third, the summary RRs were calculated to evaluate any potential difference between subsets according to the characteristics of participants.

The limitations of this study were as follows: (1) the adjusted models were different across the included studies, and these factors might have played an important role in developing vascular outcomes; (2) in a meta-analysis of published studies, publication bias was an inevitable problem; and (3) the analysis used pooled data (individual data were not available), which restricted performing a more detailed relevant analysis and obtaining more comprehensive results.

The results of this study suggested that moderate-to-severe OSA might play an important role in the risk of vascular outcomes, especially for men. Future studies should focus on specific populations to analyse the gender difference to study the association between OSA and vascular outcomes.

**Contributors**    CX carried out the studies, participated in collecting data and drafted the manuscript. RZ performed the statistical analysis and participated in its design. YT and KW helped to draft the manuscript. All authors read and approved the final manuscript.

**Funding**    This research was supported by the National Basic Research Program of China (nos. 973 Program 2015CB856405 and 2012CB720704) and the National Natural Science Foundation of China (nos. 31571149, 91432301, 81301176 and 81171273).

**Competing interests**    None declared.

**Patient consent**    Obtained.

**Provenance and peer review**    Not commissioned; externally peer reviewed.

**Data sharing statement**    No additional data available.

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
