## [Reviewer comments · BMJ Open]

ARTICLE DETAILS

TITLE (PROVISIONAL)	Association of obstructive sleep apnea with the risk of vascular outcomes and all-cause mortality: a meta-analysis
AUTHORS	Xie, Chengjuan; Zhu, Ruolin; Tian, Yanghua; Wang, Kai

VERSION 1 – REVIEW

REVIEWER	Brian Kent Guy's & St Thomas' Hospitals London United Kingdom
REVIEW RETURNED	09-Nov-2016

GENERAL COMMENTS	The authors present a meta-analysis of cardiovascular outcomes in longitudinal clinic- and community-based cohort studies of patients undergoing evaluation for sleep apnoea. The authors reasonably state that the putative relationship of OSA (and particularly severe OSA) with cardiovascular events and cardiovascular death has not been established beyond any reasonable doubt. Their manuscript is generally well written. However, I have significant concerns about their methodology/study selection. The authors state that they examined cardiovascular outcomes in >36,000 patients - unfortunately they appear to have counted many patients twice (or even three times). For example, they include two separate analyses of the Wisconsin sleep cohort study (ref 24, 26), three derived from the Sleep Heart Health Study (ref 21, 25, 27), and a further two from a longitudinal study of patients attending the Yale sleep service (ref 28,29). Each of these reports evaluated different endpoints (e.g. stroke or coronary artery disease), but each is a report from the same broad group of patients. By my rough count, this should instead leave them with a population of approx 23,000 patients. I'm not clear what the best statistical approach to this issue is. I suspect only a meta-analysis of individual patient data from these studies would allow the authors to achieve the kind of study they sought to perform here. Ultimately, I suspect that the results of such a meta-analysis would be very similar to those calculated by the authors, but I don't think the present report could survive statistical scrutiny. Additionally, I am a little concerned that this issue was not apparent to the authors. Perhaps they have taken statistical advice that this was an admissible approach to these data?
--

REVIEWER	IMRAN IFTIKHAR EMORY UNIVERSITY, USA
REVIEW RETURNED	05-Feb-2017

GENERAL COMMENTS	This is an interesting meta-analysis but I have the following primarily methodological concerns/comments:  1. The studies by Gottlieb, Campos-R, Young, Redline, Yaggi, Shah and Leao, all reported Hazard Ratio (HR). The studies by Marin and Punjabi reported data as Odds Ratio (OR). How, then did the authors of this meta-analysis derive 'relative risks' from the reported HRs and ORs from these different studies? 2. Also, in the study by Punjabi et al, data is not broken down by AHI, rather by ODI. No clarification is provided in Methods for this. 3. On a broader note, is it statistically appropriate to conduct a meta-analysis of several studies that reported outcomes in different ways (ORs in some and HRs in others)? The authors should consider reading the paper, 'Practical methods for incorporating summary time-to-event data into meta-analysis. Tierney et al, Trials 2007'. According to this paper, Odds ratios or relative risks that measure only the number of events and take no account of when they occur are appropriate for measuring dichotomous outcomes, but less appropriate for analyzing time-to-event outcomes. If the total number of events reported for each trial is used to calculate an OR or RR, this can involve combining trials reported at different stages of maturity, with variable follow up, resulting in an estimate that is both unreliable and difficult to interpret. If individual trials do not contribute data at each time point, interpretation can be misleading. Therefore, would have it have been more appropriate for the authors of this meta-analysis to simply conduct a meta-analysis of the reported HRs in each study/trial? 4. Visual inspection of funnel plots shows asymmetry contrary to what has been reported in results and discussion. All funnel plots show an assymmetrically empty left bottom corner. 5. By total mortality, do the authors mean, 'all cause-mortality'. Kindly specify. 6. Lastly, the authors should explain the interest finding of the reported increased total mortality in women as compared to men (with severe OSA) when all other such outcomes were quite the opposite (i.e., more in men compared to women)
--

VERSION 1 – AUTHOR RESPONSE

Response to reviewer #1

General comments:

Q. The authors present a meta-analysis of cardiovascular outcomes in longitudinal clinic- and community-based cohort studies of patients undergoing evaluation for sleep apnoea. The authors reasonably state that the putative relationship of OSA (and particularly severe OSA) with cardiovascular events and cardiovascular death has not been established beyond any reasonable doubt. Their manuscript is generally well written. However, I have significant concerns about their methodology/study selection.

R. We appreciate the reviewers' kind suggestions, and the details of our responses are listed as follows.

Q1. The authors state that they examined cardiovascular outcomes in >36,000 patients - unfortunately they appear to have counted many patients twice (or even three times). For example, they include two separate analyses of the Wisconsin sleep cohort study (ref 24, 26), three derived from the Sleep Heart Health Study (ref 21, 25, 27), and a further two from a longitudinal study of patients attending the Yale sleep service (ref 28,29). Each of these reports evaluated different endpoints (e.g. stroke or coronary artery disease), but each is a report from the same broad group of patients. By my rough count, this should instead leave them with a population of approx 23,000 patients.

R1. Thanks for this suggestion. We have made these changes in the revised manuscript and highlighted in RED.

Q2. I'm not clear what the best statistical approach to this issue is. I suspect only a meta-analysis of individual patient data from these studies would allow the authors to achieve the kind of study they sought to perform here.

R2. Thanks for this suggestion. In this study, prospective cohort studies that met the inclusion criteria were included, and the summary RRs with corresponding 95% CI were employed to evaluate any potential relationship between OSA and vascular outcomes or total mortality. Further, the random-effects model was then used to calculate summary RRs and 95% CIs for mild, moderate, and severe OSA versus normal as the random-effects model assumes the true underlying effect that varies among included studies. The data collected in each study was an estimate reported in the original article, and all references of statistical methods are presented in the manuscript.

Q3. Ultimately, I suspect that the results of such a meta-analysis would be very similar to those calculated by the authors, but I don't think the present report could survive statistical scrutiny. Additionally, I am a little concerned that this issue was not apparent to the authors. Perhaps they have taken statistical advice that this was an admissible approach to these data?

R3. Thanks for this suggestion. The findings of previous meta-analyses are already presented in the Introduction and Discussion sections. First, several meta-analyses have illustrated that continuous positive airway pressure (CPAP) interventions aimed at OSA may reduce the risk of cardiovascular outcomes. Kim et al. [6] showed that CPAP treatment for OSA was associated with a lower incidence of stroke and cardiac events.

Furthermore, Bratton et al. [7] indicated that among patients with OSA, use of both CPAP and mandibular advancement devices was associated with a reduction in the blood pressure. Nadeem et al. [8] suggested that CPAP treatment for OSA seemed to improve dyslipidemia (decrease in total cholesterol and low-density lipoprotein, and increase in high-density lipoprotein), whereas it did not appear to affect the triglyceride levels. These studies recommended that patients with OSA who receive interventions have a reduced risk of cardiovascular diseases. In addition, a previous meta-analysis suggested that OSA was associated with stroke, but the relationship with ischemic heart disease and cardiovascular mortality needs further research [37]. However, this study could not illustrate the impact of different levels of OSA on the risk of serious cardiovascular outcomes. Further, Dong et al. suggested that moderate-to-severe OSA significantly increased the risk of cardiovascular diseases, in particular, the risk of stroke [38]. Similarly, Ge et al. indicated that severe OSA is a strong independent predictor of cardiovascular and all-cause mortality. CPAP treatment was associated with decreased cardiovascular mortality [39]. However, these two studies could not evaluate the association of OSA with the risk of vascular outcomes and total mortality in specific subpopulations. In addition, Wang et al. suggested that severe OSA significantly increased the risk of CHD and stroke, and all-cause mortality. A positive association with CHD was observed for moderate OSA but not for mild OSA [40]. However, whether this relationship differs according to the characteristics of participants remains unclear. Finally, Xie et al. conducted a meta-analysis to evaluate the relationship between OSA and recurrent vascular events and all-cause mortality [41]. However, they just compared the highest AHI versus lowest AHI, whereas the degree of OSA and subsequent adverse outcomes were not available. Finally, in the present study, the degree of association of OSA to fatal or nonfatal CVDs, and these relationships according to different characteristics were conducted. Therefore, a comprehensive meta-analysis of these prospective cohort studies was performed to evaluate any possible correlation between OSA and vascular outcomes.

Response to reviewer #2

General comments:

Q. This is an interesting meta-analysis but I have the following primarily methodological concerns/comments:

R. We appreciate the reviewers' kind suggestions, and the details of our responses are listed as follows.

Q1. The studies by Gottlieb, Campos-R, Young, Redline, Yaggi, Shah and Leao, all reported Hazard Ratio (HR). The studies by Marin and Punjabi reported data as Odds Ratio (OR). How, then did the authors of this meta-analysis derive 'relative risks' from the reported HRs and ORs from these different studies?

R1. Thanks for this suggestion. In this study, we included prospective cohort studies that met the inclusion criteria, and hazard ratio was considered to be equivalent to RR in cohort studies. Further, given the low incidence of vascular outcomes, odds ratios (ORs) could be assumed to be accurate estimates of RRs. We have added the following sentence in the Statistical analysis section: "HR was considered to be equivalent to RR in cohort studies. Given the low incidence of vascular outcomes and all-cause mortality, ORs could be assumed to be accurate estimates of RRs."

Q2. Also, in the study by Punjabi et al, data is not broken down by AHI, rather by ODI. No clarification is provided in Methods for this.

R2. Thanks for this suggestion. We have made this change in the Statistical analysis section and highlighted in RED.

Q3. On a broader note, is it statistically appropriate to conduct a meta-analysis of several studies that reported outcomes in different ways (ORs in some and HRs in others)? The authors should consider reading the paper, 'Practical methods for incorporating summary time-to-event data into meta-analysis. Tierney et al, Trials 2007'. According to this paper, Odds ratios or relative risks that measure only the number of events and take no account of when they occur are appropriate for measuring dichotomous outcomes, but less appropriate for analyzing time-to-event outcomes. If the total number of events reported for each trial is used to calculate an OR or RR, this can involve combining trials reported at different stages of maturity, with variable follow up, resulting in an estimate that is both unreliable and difficult to interpret. If individual trials do not contribute data at each time point, interpretation can be misleading.

Therefore, would have it have been more appropriate for the authors of this meta-analysis to simply conduct a meta-analysis of the reported HRs in each study/trial?

R3. Thanks for this suggestion. In the Statistical analysis section, we have already illustrated "when more than one median of AHI levels in each study was classified into one of these three categories, the fixed-effects model was used to calculate their summary RRs and 95% CIs for effect estimates of each category" on a broader note. Further, we included prospective cohort studies that met the inclusion criteria, and hazard ratio was considered to be equivalent to RR in cohort studies. Further, given the low incidence of vascular outcomes, odds ratios (ORs) could be assumed to be accurate estimates of RRs.

Q4. Visual inspection of funnel plots shows asymmetry contrary to what has been reported in results and discussion. All funnel plots show an asymmetrically empty left bottom corner.

R4. Thanks for this suggestion. Funnel plots were employed to qualitatively evaluate the potential publication bias for MACEs, and the Egger and Begg test results were used to quantitatively evaluate any potential publication bias. If significant evidence of publication bias was detected, the trim-and-fill method was employed to adjust the publication bias.

Q5. By total mortality, do the authors mean, 'all-cause mortality'. Kindly specify.

R5. Thanks for this suggestion. Total mortality means "all-cause mortality," and we have replaced "total mortality" with "all-cause mortality" in the revised manuscript and highlighted in RED.

Q6. Lastly, the authors should explain the interest finding of the reported increased total mortality in women as compared to men (with severe OSA) when all other such outcomes were quite the opposite (i.e., more in men compared to women)

R6. Thanks for this suggestion. We have added the following sentences in the Discussion section and highlighted in RED: "No significant difference was found between mild or moderate OSA and all-cause mortality, while severe OSA was associated with an increased risk of all-cause mortality. Further, these significant associations were also observed in men and women separately. Although the effect estimate in women was larger than that in men, no gender difference was found in the relationship between OSA and all-cause mortality. The reason for this could be the number of studies that reported the relationship between severe OSA and all-cause mortality was smaller than expected, and a broad 95% CI was acquired. Therefore, the association of severe OSA with all-cause mortality in women was variable and should be verified in future large-scale prospective studies. "

VERSION 2 – REVIEW

REVIEWER	Neil Scott University of Aberdeen, UK
REVIEW RETURNED	31-May-2017

GENERAL COMMENTS	This paper contains interesting methodology to conduct meta-analyses of observational data. The paper is generally well written. One concern is the number of AHI categories used in the original studies and how well the AHI thresholds correspond to the standard categories with thresholds at 15 and 30. The authors do not provide information on how many AHI categories were used by each study. Did all the studies include multiple AHI categories or were there any that included just one category? This would mean the moderate category might include high or low risk patients. I also have some concerns about the large number of subgroup analyses. These results could be dependent on small numbers of studies with different AHI boundaries. Again, information on the AHI categories used might be useful. On the issue of combining OR, RR and HR, I think I have seen this method before – could you add a reference to reassure the reader that this methodology is robust? The “Article Summary” section (p.3) seems weak and needs changes to the wording: 1) Confidence intervals for males and females overlap and in general I don’t think this statement is justified; 2) I don’t think the phrase “statistical evidence” is clear enough; 3) and 4) The wording of these sections is not clear – please could you rewrite these. p.7: I did not fully understand the sentence on the semi-parametric method or the sentence on relative risk ratios. p.7: what is ODI? This whole sentence is not clear. In Tables 2-4, please clarify whether RRs are presented. In Tables 3-4, what method has been used to derive the results for the third line in each section? p.25: If I have, understood, I don’t think you have been able to adjust the analyses for covariates. What is meant by “adjusted” in this sentence? The search strategy does not include the alternative spelling “apnoea”. Could any studies have been missed?
---

REVIEWER	Giuseppe Biondi-Zoccai Sapienza University of Rome, Latina, Italy
REVIEW RETURNED	24-Jul-2017
GENERAL COMMENTS	All my comments have been reasonably accepted.

VERSION 2 – AUTHOR RESPONSE

Response to reviewer #3

General comments:

This paper contains interesting methodology to conduct meta-analyses of observational data. The paper is generally well written.

Response: We appreciate the reviewer's kind suggestions for improving our manuscript. Our detailed responses are provided below.

Question 1. One concern is the number of AHI categories used in the original studies and how well the AHI thresholds correspond to the standard categories with thresholds at 15 and 30. The authors do not provide information on how many AHI categories were used by each study. Did all the studies include multiple AHI categories or were there any that included just one category? This would mean the moderate category might include high or low risk patients.

Response 1: Thank you for this important question. The categories of apnea–hypopnea index (AHI) or oxygen desaturation index (ODI) in the original studies are now listed (in red font) in a new column in Table 1. For each individual study, each category of AHI was reclassified based on its calculated mid-point (for closed categories) or median (for open categories, assuming a normal distribution for AHI). The control category was composed of participants with the lowest AHI or normal participants in that study. When an individual study provided more than one median AHI level for classification among the three categories of obstructive sleep apnea (mild OSA, AHI 5–15; moderate OSA, AHI 15–30; or severe OSA, AHI > 30), a fixed-effects model was used to calculate their summary relative risks (RRs) and 95% confidence intervals (CIs) to obtain effect estimates for each category. The text in the Statistical Analysis section has been revised as follows:

“A semi-parametric method was first used to evaluate the association of mild OSA [apnea–hypopnea index (AHI): 5–15], moderate OSA (AHI: 15–30) and severe OSA (AHI > 30) with the risk of vascular outcomes or all-cause mortality in order to analyze the trend between OSA levels and vascular outcomes or all-cause mortality risk [12]. For each individual study, each category of AHI was reclassified based on its calculated mid-point (for closed categories) or median (for open categories, assuming a normal distribution for AHI). The control category was composed of participants with the lowest AHI or normal participants in that study. Furthermore, when an individual study provided more than one median AHI level for classification among the three categories (i.e. mild, moderate or severe OSA), a fixed-effects model was used to calculate their summary RRs and 95% CIs to obtain effect estimates for each category [13]. If the study data were not broken down by AHI but rather by oxygen desaturation index (ODI), classification into the OSA categories was carried out based on the judgment of the clinicians. A random-effects model was then used to calculate summary RRs and 95% CIs for mild, moderate, and severe OSA versus normal [14]. Finally, the ratio of RRs between subgroups (and the corresponding 95% CIs) were estimated using specific RRs and 95% CIs after considering the country, mean age, gender, BMI, disease status, and duration of the follow-up period [15].”

Question 2. I also have some concerns about the large number of subgroup analyses. These results could be dependent on small numbers of studies with different AHI boundaries. Again, information on the AHI categories used might be useful.

Response 2: Thank you for this query. As detailed above in Response 1, we have now provided information on the AHI categories in Table 1 and the Statistical Analysis section. We are aware that different cutoff values for AHI might affect the risk of major adverse cardiac events, and have performed our analyses according to our classification of mild, moderate and severe OSA. The findings of our subgroup analyses were based on country, mean age, gender, body mass index, disease status and duration of follow-up period according to mild, moderate, and severe OSA. These findings are presented in Table 3.

Question 3. On the issue of combining OR, RR and HR, I think I have seen this method before-could you add a reference to reassure the reader that this methodology is robust?

Response 3: Thank you for this suggestion. Numerous meta-analyses have used a similar method to combine ORs, RRs and HRs. We have now cited a previous meta-analysis (Zheng et al., 2015; new reference 11 in the manuscript) in the Statistical Analysis section to support our methodology of combining ORs, RRs and HRs.

Question 4. The “Article Summary” section (p.3) seems weak and needs changes to the wording: 1) Confidence intervals for males and females overlap and in general I don’t think this statement is justified; 2) I don’t think the phrase “statistical evidence” is clear enough; 3) and 4) The wording of these sections is not clear – please could you rewrite these.

Response 4: Thank you for this suggestion. We have revised the “Article Summary” section to read as follows:

“Article Summary:

Strengths and limitations of this study:

1. This was a meta-analysis of prospective observational studies designed to elucidate the association of obstructive sleep apnea (OSA) with fatal and nonfatal cardiovascular diseases.
2. The findings were based on a large sample size and are more robust than those obtained from any individual study.
3. The relationship was calculated for subsets of patients with specific characteristics and any potential differences between these subsets were determined.
4. Differently adjusted models might affect the progression of vascular outcomes.
5. Different cutoff values for the apnea–hypopnea index might affect the relationship between OSA and vascular outcomes.”

Question 5. p.7: I did not fully understand the sentence on the semi-parametric method or the sentence on relative risk ratios.

Response 5: Thank you for raising this query. The semi-parametric method refers to the method of reclassifying categories in individual studies so that they can be assigned to the three categories used in the present meta-analysis. The semi-parametric method we have used is described in the Statistical Analysis section as follows:

“A semi-parametric method was first used to evaluate the association of mild OSA [apnea–hypopnea index (AHI): 5–15], moderate OSA (AHI: 15–30) and severe OSA (AHI > 30) with the risk of vascular outcomes or all-cause mortality in order to analyze the trend between OSA levels and vascular outcomes or all-cause mortality risk [12]. For each individual study, each category of AHI was reclassified based on its calculated mid-point (for closed categories) or median (for open categories, assuming a normal distribution for AHI).”

Furthermore, the term “semi-parametric method” has been used in a previous meta-analysis: Li XH, Yu FF, Zhou YH, et al. Association between alcohol consumption and the risk of incident type 2 diabetes: a systematic review and dose-response meta-analysis. *Am J Clin Nutr.* 2016; 103(3): 818-29. doi: 10.3945/ajcn.115.114389.

To clarify the terminology we use in the manuscript, RR is defined as relative risk and the ratio of RRs is defined as the ratio of relative risks between subgroups. This information is provided in the Statistical Analysis section:

“Finally, the ratio of RRs between subgroups (and the corresponding 95% CIs) were estimated using specific RRs and 95% CIs.”

Question 6. p.7: what is ODI? This whole sentence is not clear.

Response 6: Thank you for pointing out this omission. We have now defined ODI as oxygen desaturation index in the Statistical Analysis section.

Question 7. In Tables 2-4, please clarify whether RRs are presented. In Tables 3-4, what method has been used to derive the results for the third line in each section?

Response 7: Thank you for this comment. Tables 2-4 present the RRs with 95% CIs and corresponding P values. A random-effects model was used to calculate summary RRs and 95% CIs for mild, moderate and severe OSA versus normal. The methodology is described in the Statistical Analysis section. Furthermore, the header row in each of Tables 2-4 has been revised (in red font) to provide this information.

Question 8. p.25: If I have, understood, I don't think you have been able to adjust the analyses for covariates. What is meant by “adjusted” in this sentence?

Response 8: Thank you for this question. We have made changes to this text in the revised manuscript. As already presented in the paragraph describing the study limitations, differences between studies in the adjusted factors might have affected the relationship between OSA and vascular outcomes.

Question 9. The search strategy does not include the alternative spelling “apnoea”. Could any studies have been missed?

Response 9: Thank you for pointing this out. We are aware that the alternative spelling “apnoea” should be employed in the search strategy. In the search tree of each database, the results of “apnoea” were contained in “apnea” due to [All fields] being used. The details of the search strategy in PubMed have already been presented in Supplemental 1.

Response to reviewer #4

General comments:

All my comments have been reasonably accepted.

Response: We appreciate the reviewer's kind comment.

VERSION 3 – REVIEW

REVIEWER	Neil Scott University of Aberdeen, UK No Competing Interest
REVIEW RETURNED	25-Sep-2017

GENERAL COMMENTS	I would like to thank the authors for making the changes to the paper. I now think the article is suitable for publication, although I still have a few minor queries. The statistical analysis section is now much clearer but what is meant by “considering” in the last sentence of the second-last paragraph? The addition of the AHI information to Table 1 is very useful. I also welcome the improvements to the headings of Tables 2-4, but shouldn’t the labels for Tables 3-4 indicate that it is a mixture of RRs and Ratios of RR? Finally, in the Abstract I think that measures of magnitude (RR (95% CI)) would be much more informative here than p-values. Also, the sentence that follows about OSA is not very clear.
--

VERSION 3 – AUTHOR RESPONSE

Response to reviewer #3

General comments:

I would like to thank the authors for making the changes to the paper. I now think the article is suitable for publication, although I still have a few minor queries.

Response: We appreciate the reviewer’s kind suggestions for improving our manuscript. Our detailed responses are provided below.

Question 1. The statistical analysis section is now much clearer but what is meant by “considering” in the last sentence of the second-last paragraph?

Response 1: Thanks for this query. We have revised this sentence to:

“Finally, the ratio of RRs and the corresponding 95% CIs between subgroups were estimated using specific RRs and 95% CIs in each group based on the country, mean age, gender, BMI, disease status, and duration of the follow-up period [15].” Further, considered have already changed into “regarded as” in the last sentence of Statistical analysis.

Question 2. The addition of the AHI information to Table 1 is very useful. I also welcome the improvements to the headings of Tables 2-4, but shouldn’t the labels for Tables 3-4 indicate that it is a mixture of RRs and Ratios of RR?

Response 2: Thanks for this suggestion. We have added labels to Tables 3 and 4 accordingly.

Question 3. Finally, in the Abstract I think that measures of magnitude (RR (95% CI)) would be much more informative here than p-values. Also, the sentence that follows about OSA is not very clear.

Response 3: Thanks for this suggestion. We have made these changes in the revised manuscript, which are marked “RED”.

VERSION 4 – REVIEW

REVIEWER	Neil Scott University of Aberdeen, UK
REVIEW RETURNED	10-Oct-2017
GENERAL COMMENTS	I thank the authors for implementing the latest suggestions into the paper. I am happy with the changes made and the paper is now suitable to be published.